# Deep Sequencing of Immunoglobulin Genes Identifies a Very Low Percentage of Monoclonal B Cells in Primary Cutaneous Marginal Zone Lymphomas with CD30-Positive Hodgkin/Reed–Sternberg-like Cells

**DOI:** 10.3390/diagnostics12020290

**Published:** 2022-01-24

**Authors:** Arianna Di Napoli, Evelina Rogges, Niccolò Noccioli, Anna Gazzola, Gianluca Lopez, Severino Persechino, Rita Mancini, Elena Sabattini

**Affiliations:** 1Department of Clinical and Molecular Medicine, Sant’Andrea Hospital, Sapienza University, 00189 Rome, Italy; evelina.rogges@uniroma1.it (E.R.); niccolo.noccioli@uniroma1.it (N.N.); gianluca.lopez10@gmail.com (G.L.); rita.mancini@uniroma1.it (R.M.); 2Haematopathology Unit, IRCCS Azienda Ospedaliero Universitaria di Bologna, 40138 Bologna, Italy; gazzola.anna1981@gmail.com (A.G.); elena.sabattini@aosp.bo.it (E.S.); 3NESMOS Department, Dermatology Unit, Sant’Andrea Hospital, Sapienza University, 00189 Rome, Italy; severino.persechino@uniroma1.it

**Keywords:** cutaneous marginal zone lymphoma, Hodgkin and Reed–Stemberg cells, IGH and IGK rearrangements, NGS

## Abstract

The spectrum of cutaneous CD30-positive lymphoproliferative disorders encompasses both inflammatory and neoplastic conditions. CD30+ Hodgkin and Reed–Sternberg-like cells have been occasionally reported in primary cutaneous marginal zone lymphoma, where they are thought to represent a side neoplastic component within a dominant background of lymphomatous small B cells. Herein, we describe the histological and molecular findings of three cases of primary cutaneous marginal zone lymphomas with CD30+ H/RS cells, in which next-generation sequencing analysis revealed the clonal population to consist in less than 5% of the cutaneous B-cell infiltrate, providing a thought-provoking focus on a possible main role for CD30+ cells in primary cutaneous marginal zone lymphoproliferations.

## 1. Introduction

Primary cutaneous marginal zone lymphomas represent approximately 2–7% of all cutaneous lymphomas. They typically occur in adults with male predominance and present red to violaceous papules, nodules, or plaques on the extremities and trunk without evidence of extracutaneous involvement [1,2,3,4,5,6,7,8,9,10,11]. Histologically, PCMZLs are characterized by a dense nodular and variably diffuse dermal perivascular and periappendageal infiltrate of polymorphous small to medium-sized B cells, which spares the epidermis with a Grenz zone, and occasionally involves subcutaneous fat. Plasma cells with monotypic expression of immunoglobulin light chains are also frequently observed. Follicles with reactive germinal centers may be present and colonized by the neoplastic marginal zone cells and plasma cells [2,11,12,13,14,15]. 

The 2018 updated WHO-EORTC classification for primary cutaneous lymphomas recognizes two different subtypes of PCMZL based on the expression of class-switched immunoglobulins and the chemokine receptor CXCR3 [1,2,16,17]. The class-switched cases are composed of IgG-positive, IgA-positive, or IgE-positive and CXCR3-negative B cells admixed with numerous reactive T-cells and peripherally clustered monotypic plasma cells. Due to the lack of colonization of reactive germinal centers by neoplastic B cells, lymphoepithelial lesions, or transformation into a diffuse large B-cell lymphoma, these cases are regarded by some authors as clonal chronic cutaneous lymphoproliferative disorders (LPD) rather than overt lymphomas [1,2,11,16,17,18,19,20]. A monoclonal rearrangement of immunoglobulin heavy (IGH) or light chain (IGK/IGL) genes can help in distinguishing PCMZL from cutaneous lymphoid hyperplasia (B-cell pseudolymphoma), which can histologically mimic PCMZL. However, clonal IGH or IGK rearrangements have been seen in benign cutaneous lymphoid proliferations, hence the importance of integrating clinical and molecular data with histopathologic and immunophenotypic features [19,20,21,22,23]. 

The non-class-switched cases are instead considered true lymphomas consisting of large nodules of neoplastic B cells expressing IgM and CXCR3 with scattered plasma cells and a less prominent T-cell infiltrate.

Nevertheless, PCMZLs have an indolent clinical course (5-year disease-specific survival rate close to 100%) [1,4,6,7,8,24] with frequent cutaneous recurrences (36% to 71% of patients) [4,6,7,8] but infrequent transformation into diffuse large B-cell lymphoma (DLBCL) [4,5,6,7,8,9,11,24,25]. Although some cases of PCMZL show increased numbers of scattered large B cells, transformation into a DLBCL is generally recognized by the presence of solid sheets of large transformed B cells. Interestingly, in rare cases of PCMZLs with CD30-positive Hodgkin and Reed–Sternberg-like (H/RS) cells reported in the literature, the number of CD30+ H/RS-like cells has been associated with a more advanced clinical stage and multiple relapses of the disease [26].

Herein, we describe three cases of PCMZL with CD30+ H/RS cells, which were investigated by using histopathology and genetic analysis of immunoglobulin gene rearrangement with sub-clonal resolution in an attempt to clarify the possible impact of the CD30-positive component.

## 2. Materials and Methods

### 2.1. Patients, Tissues Samples

Three cases of cutaneous lymphoproliferative disorders with CD30+ H/RS-like cells, formalin-fixed paraffin-embedded (FFPE), were selected from the files of the Department of Pathology, La Sapienza University, Rome, and the Department of Hematopathology, S. Orsola University Hospital, Bologna. In all the cases, the diagnosis of PCMZL was performed primarily on skin localization in the absence of previous or concurrent involvement of any other extra-nodal or nodal site. All of the cases were reviewed by expert hematopathologists by utilizing morphological and immunohistochemical criteria according to WHO and EORTC classifications. 

### 2.2. Immunohistochemistry and In Situ Hybridization

Paraffin sections were immunostained for CD30, CD3, CD5, CD20, CD79a, CD15, CD10, CD21, CD23, BCL2, BCL6, Ki-67, OCT2, MUM1, kappa, lambda, LMP1, PD1, and PAX5 (Dako, Agilent, Santa Clara, CA, USA) using an automated immunostainer (Dako Omnis, Agilent, Santa Clara, CA, USA). In situ hybridization for EBER was performed on paraffin sections using Epstein–Barr Virus (EBER), PNA Probe/Fluorescein, and FITC/HRP (Dako, Agilent, Santa Clara, CA, USA). 

### 2.3. Molecular Analyses of Immunoglobulin Heavy (IGH) and Light Chain (IGK) Gene Rearrangements

The DNA extracted from paraffin-embedded samples was used to assess immunoglobulin heavy (IGH) and light chain (IGK) gene rearrangements using both a gene scan polymerase chain reaction (PCR) approach (IdentiClone, Invivoscribe Inc., San Diego, CA, USA) and the next-generation sequencing (NGS) LymphoTrack Dx IGH and IGK assays (Invivoscribe, Inc., San Diego, CA, USA), as previously described [27,28]. Briefly, gene scan analysis was conducted using a multiplex PCR, developed within the European BIOMED-2/EuroClonality consortium, followed by size analysis of fluorescently labeled PCR fragments separated by capillary electrophoresis. NGS analysis consisted of amplification by PCR of multiple master mixes containing primers designed with barcoded sequence adaptors. After purification and quantification, libraries were sequenced on an Ion PGM^TM^ instrument (Thermo Fisher, Waltham, MA, USA). The FASTQ files generated were analyzed with LymphoTrack PGM software (version 2.3.1). To determine V, (D), and J genes segments usage of IGH and IGK locus, sequences were submitted to the international ImMunoGeneTics database (IMGT) and aligned to the closest matching germline gene by using the IMGT/V-QUEST and IMGT/Junction Analysis software (http://www.imgt.org/IMGT_vquest/input) (accessed on 10 July 2020) (version 3.5.25). IGH and IGK clonal and polyclonal controls were included for both gene scan and NGS analyses (Invivoscribe, Inc.). Both the LymphoTrack-PGM software and the IMGT/V-QUEST and IMGT/Junction Analysis softwares consider an Ig rearrangement as nonproductive when no complete heavy or light Ig chain can be produced due to the presence of stop codons caused by deletions, mutations, or frameshift mutations.

## 3. Results

### 3.1. Case 1

A 67-year-old patient with a reddish elevated nodule of 2 cm diameter on the right thigh underwent an excisional biopsy in February 2013. The histological examination showed a diffuse full-thickness lymphoid infiltrate involving subcutaneous soft tissue. A narrow Grenz zone of uninvolved dermis separated the lymphoid infiltrate from the epidermis. The infiltrate was composed of CD20+, CD79a+, PAX5+ CD10-, BCL6-, BCL2+, CD23+, CD5-, CyclinD1-, MUM1-, IgM+, IgD+, IgG-, and IgA-small B cells, admixed with abundant small CD3+ and CD5+ T-cells, and plasma cells with prevalent lambda light chain immunoglobulin expression. Immunohistochemistry for both CD21 and CD23 did not highlight a meshwork of follicular dendritic cells. Scattered large proliferating cells with H/RS morphology were also present. These large cells expressed CD30, CD15, BCL6, and MUM1, stained weakly for PAX5 and occasionally for CD20 and CD79a, and they were associated with CD3+, CD5+, and PD-1+ T-cell rosettes. Epstein–Barr virus (EBV) infection was ruled out by determining negativity both for in situ hybridization for EBV-encoded RNA transcripts (EBER), and for immunohistochemistry for EBV latent membrane protein 1. The proliferation index was around 10%, and Ki-67 immunoreactivity was mainly associated with large atypical cells (Figure 1A–I). Based on morphological and immunophenotypical findings, a diagnosis of PCMZL with H/RS-like cells was made. PET and CT scans excluded any other localization of the disease, and a watch-and-wait strategy was adopted.

Five years later, a 3 cm cutaneous nodule appeared adjacent to the scar of the original lesion on the right thigh. An excisional biopsy was performed. The histology revealed a diffuse infiltrate of a monomorphic population of medium/large-sized cells extending throughout the dermis, without involvement of the superficial epidermis. Immunohistochemistry showed positivity of the neoplastic cells for CD20, CD79a, CD30, and BCL6, and negativity for CD15, CD10, CD23, BCL2, MUM1, IgM, Kappa, Lambda, LMP1, and EBER. MYC was expressed in about 30% of tumor cells. The proliferation index was high (Ki-67 = 70%) (Figure 1J–R). The lesion was diagnosed as CD30+ primary cutaneous diffuse large B-cell lymphoma not otherwise specified (CD30+ PCDLBCL-NOS). PET and CT scans excluded any other localization of the disease, and a watchful waiting strategy was adopted.

In order to evaluate whether CD30+ PCDLBL represented the clonal evolution of PCMZL, we performed a comparative analysis of IGH gene rearrangement using both multiplex polymerase chain reaction (PCR) gene scan analysis and next-generation sequencing (NGS). Gene scan analysis revealed an identical peak at 263bp within a polyclonal background in both lesions. This corresponded to a productive IGH rearrangement (VH1-2_03- JH6_02) that accounted for 61.82% of the total reads in the CD30+ PCDLBCL-NOS (total reads count 15,358) and for only 4.55% of the total reads in the PCMZL specimen (total reads count 224,588) (Figure 2). A clonotype is defined if the merged sequences of the rearranged Ig are encoded by the same VH/JH, Vκ/Jκ, or Vλ/Jλ gene segments and possess an identical amino acid sequence and length in the third complementarity-determining region (CDR3), which corresponds to the junction of the V-(D)-J segments with the addition of nucleotides at the joints. Both lesions of patient 1 showed the same usage of VH/JH gene segments and had an almost identical third complementarity-determining region (CDR3) (Figure 2D,H). Moreover, they shared the vast majority of the IGVH somatic hypermutations (Appendix A). These findings support a clonal relationship between PCMZL with H/RS-like cells and PCDLBCL. The presence of the clonotype in only 4.55% of the total reads in the PCMZL was quite surprising; the majority of the B-cell component was expected to be neoplastic, since reactive polyclonal follicles within the lesion were absent. A PCR-based analysis of IGK gene rearrangement was also performed, and the results overlapped with those obtained for the IGH gene. Indeed, in the PCMZL with H/RS-like cells, the presence of a smaller clonal peak of the same size as the one detected in the PCDLBCL sample was observed (Appendix A). 

### 3.2. Case 2 

A 23-year-old patient with a cutaneous nodule on the left arm underwent an excisional biopsy in 2017. The histology showed a diffuse lymphocytic infiltrate involving the deep dermis and the adipose subcutaneous tissue composed of some reactive follicles (CD10+, BCL6+, and BCL2-) with an expanded IgM+, IgD+, and IgG- marginal zone, surrounded by small T CD3+ lymphocytes admixed with lambda light chain monotypic plasma cells and scattered large H/RS-like cells. The H/RS-like component expressed CD30, CD15, PAX5, and OCT2, stained focally for CD20, CD79a, and BCL6, and was negative for LMP1, EBER, and BCL2. Rosettes of CD3+ PD-1+ T-cells surrounded the large atypical cells (Figure 3A–G). 

PCR clonality testing detected a prominent IGH clonal peak that corresponded to a productive VH3-15_02-JH6_02 rearrangement by NGS analysis accounting for 5.81% of the total reads (total reads count 9977) (Figure 4A–C). A final diagnosis of PCMZL with H/RS CD30+ large cells was made. Staging CT and PET scans were negative. 

### 3.3. Case 3

A 34-year-old patient underwent an excisional biopsy of a single cutaneous papule on the right leg in 2019. The histology showed, in the upper and lower dermis, a nodular infiltrate of small-lymphocytes partly organized in the formation of B-cell follicles with CD20+, PAX5+, IgM+, IgD+, and IgG- expanded marginal zones, admixed with small CD3+ and PD1+ T-cells rosetting large H/RS-like cells with a CD30+, CD15+,CD20+, PAX5+, CD79a+, OCT2+, BCL6+/-, EMA-, and EBER- phenotype. Plasma cells with lambda light chain restriction were also present (Figure 3H–N). 

Molecular analyses showed a polyclonal IGH rearrangement and an unproductive VK2D-29_01-JK4_01 and a deletional IGKINTRON-IGKDEL clonal rearrangement of the IGK gene detected in 4.55% and 1.66% of the total reads, respectively (total reads count 67,771) (Figure 4D–G). A conclusive diagnosis of PCMZL with H/RS-like cells was made. No other localization of the disease was detected by CT and PET scans.

## 4. Discussion

CD30 is a transmembrane cytokine receptor belonging to the tumor necrosis factor (TNF) receptor superfamily thought to regulate cell proliferation, differentiation, and apoptosis [29,30]. Although it was originally identified on the surface of Hodgkin and Reed–Sternberg cells in patients with Hodgkin lymphoma [31], it may be expressed by T cells or B cells in different benign and neoplastic conditions and can be induced in vitro by mitogens or viruses [30]. In skin lesions, CD30-positive lymphocytes can be detected in persistent arthropod bites, cutaneous herpes virus infections, and drug reactions [32,33,34,35]. Among malignancies, CD30-positive T-cells typically characterize lymphomatoid papulosis (LyP) and primary cutaneous anaplastic large cell lymphoma (cALCL) [1,36], whereas in B-cell neoplasms CD30-positive cells are less common, occurring in the setting of iatrogenic immunosuppression (i.e., mucocutaneous ulcers) [1,37], and in primary cutaneous diffuse large B-cell lymphomas (DLBCL) [38]. CD30 expression has also been reported in a variable number of neoplastic cells of primary cutaneous follicle center lymphoma (PCFCL) and of primary cutaneous marginal zone lymphoma (PCMZL) [14,15,39,40,41,42], with rare cases describing a Hodgkin and Reed–Sternberg (H/RS)-like morphology [26,43,44]. 

In PCFCL, a high percentage of CD30+ cells (about 70%) was not associated with sheets of centroblasts, excluding PCDLBCL diagnosis. In contrast, in PCMZL, the presence of more than 15% of CD30+ H/RS-like cells has been associated with a more advanced clinical stage and multiple relapses of the disease [26]. Here, we reported three cases of PCMZL with CD30+ H/RS-like cells, one of which relapsed as CD30+ PCDLBCL 5 years later. The percentage of CD30+ cells in all three cases was low, accounting for less than 5% of the B cells. Of note, NGS analysis of heavy and light chain genes revealed a clonal rearrangement in 4% to 6% of total reads, suggesting that the neoplastic population is only a small minority of the B-cell cutaneous infiltrate. In contrast, in the CD30+ cutaneous relapse, the percentage of the clonotype identified, which was 4.55% at the time of the initial diagnosis, increased to 61.82%. In the skin, similar molecular results have been reported in the possible progression of LyP to cALCL [45,46,47,48,49,50,51] and of B-cell pseudolymphoma to cutaneous B-cell lymphoma [19,20,21,22,23]. In both situations, it has been suggested that chronic inflammation plays a major role by favoring the accumulation of somatic mutations that induce the transition from a polyclonal to a monoclonal lymphoproliferation and then the progression to an overt lymphoma. Although a unifying etiologic factor has not been identified in patients with PCMZL, evidence suggests that at least one subset of these lymphomas is the result of a chronic antigenic stimulation, with Borrelia Burgdorferi infection detected mainly in European patients [52,53,54]. We may suppose that PCMZL may arise in the context of chronic inflammation in which an initial expansion of a polyclonal B-cell population progresses toward a monoclonal expansion. This hypothesis is shared for other types of MZL, such as those arising in patients with Sjogren syndrome and HCV-infection [55,56,57]. Although our findings are limited by the impossibility of performing microdissection of CD30+ H/RS-like cells, since the bioptic tissue has been already extensively used for the immunohistochemical and molecular analyses, they prompt us to speculate that in PCMZL CD30+ cells may represent an important component of the expanding clonal B-cell population. Similarly, Prieto-Torres and collaborators found a progressive increase in the number of CD30+ cells in the relapse of two cases of PCMZL with H/RS-like cells, suggesting that CD30 may indicate a greater tendency for lesions to recur [26]. Indeed, although all of our three cases exhibited the presence of plasma cells with a prevalent lambda light chain expression at the time of diagnosis, relapsed tumor cells did not show cytoplasmic light chain immunoglobulin expression, but they retained the expression of CD30. Furthermore, it is known that monotypic light chain restriction is not equivalent to clonality, as reported in atypical marginal zone hyperplasia of mucosa-associated lymphoid tissue of childhood [58].

The presence of only a small clonal B-cell population may raise concerns about the diagnosis of cutaneous lymphoma over a clonal chronic lymphoproliferative disorder. However, based on the 2018 updated WHO-EORTC classification, true PCMZL is characterized by IgM and IgD non-class-switched immunoglobulins [1]), which were actually expressed in all of our three cases. The possibility of a secondary cutaneous involvement by a nodal classical Hodgkin lymphoma (CHL) was also ruled out by the negativity of both CT and PET scans in our patients. A very small number of cases of CHL have also been reported to be restricted to the skin without evidence of systemic involvement [59,60]. However, a diagnosis of primary cutaneous CHL was not favored in our patients because the background was not composed mainly of T cells, macrophages, and eosinophils, as in CHL, but of B cells diffusely infiltrating the dermis or mainly organized in follicles with expanded marginal zones and monotypic plasma cells. In addition, the first case relapsed as a DLBCL, a transformation more commonly seen in nodular lymphocyte-predominant HL than in CHL. Similarly, a diagnosis of an EBV-related CD30+ lymphoproliferative disorder of the skin, such as the mucocutaneous ulcer, was ruled out due to EBER negativity and the absence of iatrogenic immunosuppression.

In marginal zone lymphomas, it is not unusual to find a prominent clonal peak within a polyclonal background when utilizing multiplex PCR. However, based on the height of the clonal peaks and the abundance of the cutaneous B-cell infiltrates, it was rather unexpected to detect such a low percentage of the clonotypes by NGS analysis in all of our three cases. Although PCR-based and NGS-based clonality testing are reported to be highly concordant, a linear equivalence between the height of the peak revealed by gene scan analysis and the percentage of the clonotypes detected using NGS could not be determined [27,28,61]. This is because the PCR fragment assay is less specific than the nucleic acid sequence, since it only considers the size in base pairs of the PCR amplicon, which may be the same for multiple PCR fragments. Next-generation sequencing (NGS) technology significantly improves clonality testing not only by providing data on the VDJ rearranged sequence, thus enabling comparisons across different biopsy sites and timepoints, but also by increasing sensitive detections of clonal populations providing their relative percentages of the total merged reads [27,28,61]. There are two possible interpretive guidelines for the NGS clonality assays: (1) a clonal rearrangement is defined by the presence of a specific clonotype in ≥2.5% of total reads of the merged rearrangement sequences; and (2) a clonal rearrangement should be ≥3 times the percentage reads of the third top-merged sequence [28]. In all of our three cases, the clonal sequence represented ≥2.5% of total reads and was ≥3 times higher than the third top-merged sequence; thus, both criteria were satisfied.

In conclusion, we reported three PCMZLs with H/RS-like cells in which the clonotype, quantified by NGS-based clonality testing, accounted for a low percentage (about 5%) of the B-cell population. Further studies are warranted in order to clarify the possible relationship between the presence of CD30+ H/RS-like cells and the molecular aspects of PCMZL.

## Figures and Tables

**Figure 1 diagnostics-12-00290-f001:**
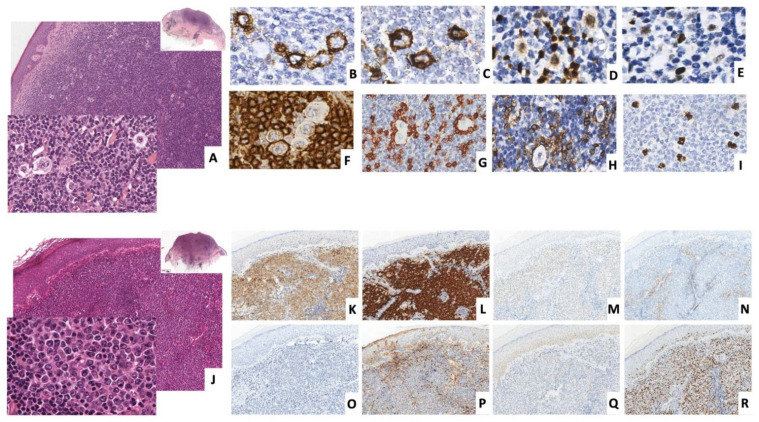
Histology of the skin lesions of patient 1 at diagnosis (**A**–**I**) and at relapse (**J**–**R**). At diagnosis the dermis was infiltrated by a small cell lymphocytic infiltrate with several large mono-nucleated or bi-nucleated cells (H/RS-like morphology). ((**A**), hematoxylin and eosin, 100×, upper insert 20×, lower insert 400×) expressing CD30 ((**B**), 400×), CD15 ((**C**), 400×), PAX5 ((**D**), 400×), BCL6 ((**E**), 400×), and occasionally CD20 ((**F**), 400×), and rosetting with small reactive CD3-positive ((**G**), 400×) and PD1-positive ((**H**), 400×) T lymphocytes. The proliferation index was low and mainly restricted to large cells ((**I**), 400×). The cutaneous lesion, relapsed on the scar of the previous nodule, showed a diffuse dermic infiltrate of monomorphic medium-sized to large-sized atypical lymphoid cells ((**J**) hematoxylin and eosin, 100× upper insert 20×, lower insert 400×), positive for CD30 ((**K**), 100×), CD79a ((**L**), 100×), and BCL6 ((**M**), 100×) and negative for CD10 ((**N**), 100×), BCL2 ((**O**), 100×), and IgM ((**P**), 100×). MYC was positive in about 30% of the cells ((**Q**), 100×). The proliferation index (Ki67) was high ((**R**), 100×).

**Figure 2 diagnostics-12-00290-f002:**
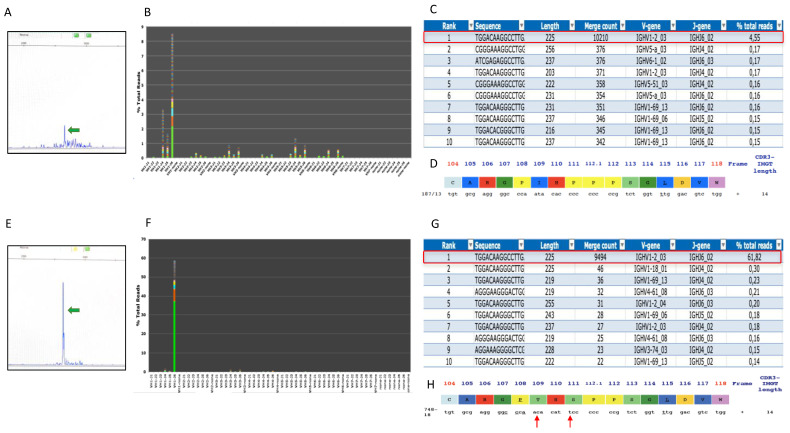
IGH gene rearrangement of the skin lesions of patient 1 at diagnosis (**A**–**C**) and at relapse (**E**–**G**) by PCR-based (**A**,**E**) and NGS-based (**B**,**C**,**F**,**G**) methods. Gene scan analysis of multiplex PCR analysis revealed in framework region 2 (FR2) a clonal peak (green arrows) with an identical size in both lesions (**A**,**E**). Snapshot of the V-J sequence frequency graph of LymphoTrack PGM Analysis Software showed in FR2 two clonal sequences with the same VH/JH gene segment use (VH1-J6) in both (**B**,**F**). The merged read summary output showed that the same clonotype accounted for 4.55% of total reads at diagnosis (**C**) and for 61.82% of total reads at relapse (**G**). Using the IMGT/V-QUEST program, both IGH rearrangements came out as productive with CDR3 regions of identical length and almost the same sequence (**D**,**H**) except for 2 nucleotides (red arrows) encoding for 2 different amino acids (letters in the colored bar).

**Figure 3 diagnostics-12-00290-f003:**
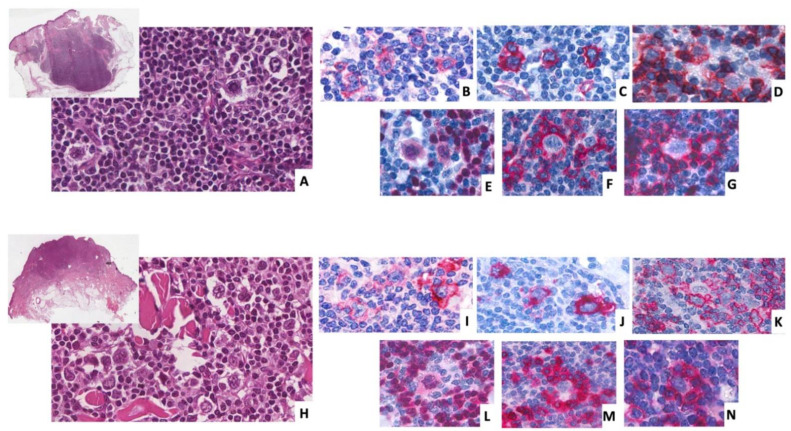
Histology of cutaneous lesion of patient 2 (**A**–**G**) and of patient 3 (**H**–**N**). Patient 2 showed a diffuse lymphocytic infiltrate involving the dermis and the adipose subcutaneous tissue, whereas patient 3 showed a nodular infiltrate confined to the dermis. In both cases, the infiltrate was mainly composed of small lymphocytes with sparse plasma cells and scattered large cells with H/RS-like morphology ((**A**,**H**) hematoxylin and eosin (**H**,**E**) 400×, upper insert 20×). The large atypical cells were strongly positive for CD30 ((**B**,**I**), 400×) and CD15 ((**C**,**J**), 400×), and showed a heterogeneous expression of CD20 ((**D**,**K**), 400×) and PAX5 ((**E**,**L**) 400×). Small reactive CD3+ ((**F**,**M**), 400×) and PD1+ ((**G**,**N)**, 400×) T cells were rosetting around the H/RS-like cells.

**Figure 4 diagnostics-12-00290-f004:**
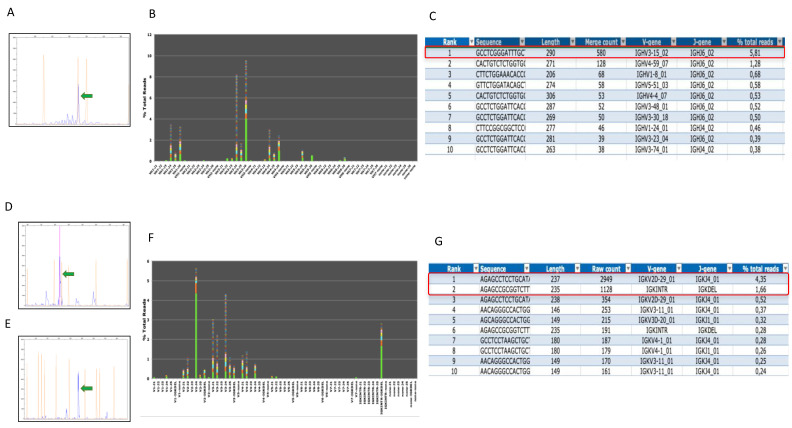
IGH gene rearrangement of the skin lesion of patient 2 (**A**–**C**) and IGK gene rearrangement of the skin lesion of patient 3 (**D**–**G**) by PCR-based (**A**,**D**,**E**) and NGS-based (**B**,**C**,**F**,**G**) methods. In patient 2, a predominant peak over a polyclonal IGH rearrangement observed by PCR ((**A**), green arrow) was confirmed by NGS analysis using FR1 primer sets. A productive VH3-J6 clone was detected in 4% of total reads as shown in the merged read summary graph (**B**) and output table (**C**). In patient 3, IGK assay demonstrated two dominant peaks ((**D**), green arrows). One was obtained using the primers amplifying the Vk-Jk region and corresponded to an unproductive VK2-J4 rearrangement (**F**) detected by NGS analysis in 4.35% of total reads (**G**). The other peak was observed using the primers covering the Vk/Jk-Ck (INTR)-kappa deleting element (Kde) region (**E**) and corresponded to an IGKINTRON-IGKDEL sequence by NGS (**F**) that accounted for 1.66% of total reads (**G**).

## Data Availability

The authors confirm that the data supporting the findings of this study are available within the article and its Appendix A.

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
