# Peer review of "Deep Sequencing of Immunoglobulin Genes Identifies a Very Low Percentage of Monoclonal B Cells in Primary Cutaneous Marginal Zone Lymphomas with CD30-Positive Hodgkin/Reed–Sternberg-like Cells"

_diagnostics, 2022, doi:10.3390/diagnostics12020290_

Round 1

Reviewer 1 Report

The paper is simple and easy to follow. It present 2 main types of analyses with three patients.

The introduction is very well written, but deserves to be longer and more precise. Details are lacking.

The material and method part is short but sufficient.

To be sure ‘Histology of the skin lesions of patient 1 at diagnosis (upper line) and at relapse (lower line)’.  Does i means ‘Histology of the skin lesions of patient 1 at diagnosis (upper line, label A to I) and at relapse (lower line, label J to R)’.  If yes, please add in the legend.

Similarly, why was it x400 fpr B to I and x 100 for the others?

In terms of quantification on the Figure 1, are all markers really significant? Reader has the impression that some markers as CD30 is low in B in regards to K.

For Figure 2, I was wondering the quantitative value we can have for the quality of the sequencing? Is it really robust?

There is Legend error fo PCR (A, E, F) is it not (A,D, F)

How is defined unproductive rearrangement? And confirmed?

Author Response

POINT1:The introduction is very well written, but deserves to be longer and more precise. Details are lacking. The material and method part is short but sufficient.

RESPONSE 1: We agree with the Reviewer, hence the introduction has been integrated with additional details about the histopathology of PCMZL and the role of clonality analyses in its diagnosis.

POINT 2 :To be sure ‘Histology of the skin lesions of patient 1 at diagnosis (upper line) and at relapse (lower line)’.  Does i means ‘Histology of the skin lesions of patient 1 at diagnosis (upper line, label A to I) and at relapse (lower line, label J to R)’.  If yes, please add in the legend.

RESPONSE 2: It was now added in the figure legend.

POINT 3: Similarly, why was it x400 fpr B to I and x 100 for the others?

RESPONSE 3: The magnification x400 was used to show the H/RS cells, whereas a lower magnification (x100) was preferred to demonstrate the diffuse staining for both CD30 and CD20 in the relapsed lesion that otherwise in the x400 images would not be appreciated.

POINT 4: In terms of quantification on the Figure 1, are all markers really significant? Reader has the impression that some markers as CD30 is low in B in regards to K.

RESPONSE 4: In the relapse the infiltrate was completely different from the first biopsy except for the retained expression of CD30. In particular, at diagnosis CD30 (B) highlights the presence of H/RS-like cells intermingled with numerous CD20+ small B cells (F). whereas, in the relapse the infiltrate was composed of large CD20+ B cells (L) that also retain expression of CD30 (K). The additional immunohistochemical markers were shown to demonstrate in the first biopsy the similarity of the of H/RS-cells with those usually seen in Hodgkin lymphoma (C-I), and at relapse to exclude other diagnosis such as primary cutaneous follicle centre lymphoma and DLBCL Leg-type (M-R).

POINT 5: For Figure 2, I was wondering the quantitative value we can have for the quality of the sequencing? Is it really robust?

RESPONSE 5: Sensitivity of next-generation sequencing (NGS)-based clonality assessment on lymphoid tissue and lymphoma specimens has been addressed by Scheijen et al in their report (Scheijen, B., Meijers, R.W.J., Rijntjes, J. et al. Next-generation sequencing of immunoglobulin gene rearrangements for clonality assessment: a technical feasibility study by EuroClonality-NGS. Leukemia 33, 2227–2240 (2019). In particular, the abundance of IG gene rearrangements and the relative distribution of clonotypes was analyzed using decreasing concentrations of Qubit-quantified input DNA for each multiplex PCR reaction, ranging from 40 to 1.25 ng, of formalin-fixed paraffin-embedded (FFPE) specimens of two clonal B cell lymphoma samples. Clonal rearrangements could still be detected at the lowest input concentration of 1.25 ng, indicating that NGS can detect rearrangements at this low DNA concentration in case of a clonal B cell lymphoma. Moreover, they analyzed four B cell lymphoma specimens with a high tumor load (80–90%) in a dilution series at concentrations of 10%, 5%, 2.5%, and 1% in a polyclonal background of FFPE tonsil DNA, using a fixed amount of total input DNA (40 ng). The data for IGHV-IGHD-IGHJ, IGHD-IGHJ, and IGKV-IGKJ gene rearrangements demonstrated that, in the majority of cases, clonal rearrangements could be traced back at 10%, 5%, and 2.5% dilutions and in two cases the clonotype for IGHD-IGHJ and IGKV-IGKJ gene rearrangements could even be detected at 1.25 ng input DNA. Additionally, they demonstrated that there was a high reproducibility in the identification of clonotypes in polyclonal tonsil specimens between frozen and FFPE tissues, and also among different centres.

POINT 6: There is Legend error fo PCR (A, E, F) is it not (A,D, F)

RESPONSE 6: It has been corrected in the figure legend.

POINT 7: How is defined unproductive rearrangement? And confirmed?

RESPONSE 7: An unproductive Ig rearrangement is defined when no complete heavy or light Ig chain can be produced due to deletions, mutations or frame shift mutations that lead to stop codons. This may be attributable to the splicing process between V-(D)-J gene segments, which could be imprecise. This concept has been now explained in the molecular analyses section of the material and methods. Except for NGS, only Sanger sequencing of the rearranged Ig could confirm if it is productive or not.

Reviewer 2 Report

This is a nice and interesting study with very beautiful illustration of the histological and immunohistological finding. In order to assess the molecular relationship for case 1, IgH clonality analysis was performed by a commercial NGS assay for IgH and – for two cases – IgL Kappa amplification. A very convincing B-cell clonality was described in the relapse lesion which was also found to much lower extent in the primary cutaneous lesion. However, for both lesions, apparently identical or almost identical sequences (column “sequence” in the table of figure 2) with different length. This unusual finding requires very clear explanation including a description what this “sequence” is representing. For the second case there is a CD30 positive lesion at only one time point which was analyzed for IgH and IgL-Kappa by NGS. For both, the most dominant clonotype is around 4% although the visual impression indicates a much higher percentage – at least for IgL kappa. I fully agree with authors that this dominant Ig rearrangement might derive from the CD30-positive cell population. However, a formal proof is missing. For case 3 there is unfortunately no graphical representation of the results, but a similar finding as for case 2, raising the same comment as for case 2.

In contrast to the histological/immunohistological description, the molecular findings were less precisely described and unclear to some extent. As already mentioned above, it is unclear what the “sequence”  in the tables for case 2 and 3 is representing. It is also not described which IgH framework region is amplified. A simple reference to the vendor is in this case not sufficient for a scientific publication. Furthermore, the range of the amplicon sizes (IgH) appears to be very huge (203 to 306 bp). Why was IgL not performed for case 1; it should be able to support the IgH findings. Finally, I assume that – based on length of the amplicons – somatic hypermutation (SHM) should be analyzable. For case 1, is SHM identical at both time points?  

Author Response

POINT 1: This is a nice and interesting study with very beautiful illustration of the histological and immunohistological finding. In order to assess the molecular relationship for case 1, IgH clonality analysis was performed by a commercial NGS assay for IgH and – for two cases – IgL Kappa amplification. A very convincing B-cell clonality was described in the relapse lesion which was also found to much lower extent in the primary cutaneous lesion. However, for both lesions, apparently identical or almost identical sequences (column “sequence” in the table of figure 2) with different length. This unusual finding requires very clear explanation including a description what this “sequence” is representing.

RESPONSE 1: A clonotype is defined if the merged sequences of the rearranged Ig are encoded by the same VH/JH, Vκ/Jκ or Vλ/Jλ gene segments and possess an identical amino acids sequence and lenght in the third complementarity determining region (CDR3). Both lesions of patients 1, showed the same VH/JH gene segments usage (now reported also in the supplementary figure 1) and had an almost identical third complementarity determining region (CDR3), which corresponds to the junction of the V-D-J segments with the addition of nucleotides at the IGHV-IGHD and IGHD-IGHJ joints. (Figure 2 D, H). Moreover, they shared the vast majority of the IGVH somatic hypermutations (supplementary Figure 1). These findings support a clonal relationship between the PCMZL with H/RS-like cells and the PCDLBCL. This has been now better explained in the text.

POINT 2: For the second case there is a CD30 positive lesion at only one time point which was analyzed for IgH and IgL-Kappa by NGS. For both, the most dominant clonotype is around 4% although the visual impression indicates a much higher percentage – at least for IgL kappa. I fully agree with authors that this dominant Ig rearrangement might derive from the CD30-positive cell population. However, a formal proof is missing.

RESPONSE 2: The molecular analyses showed in Figure 4 refer to IGH sequencing of case 2 and to IGK sequencing of case 3. For patient 3 we indicated the IGK sequencing only because of the presence of a polyclonal rearrangement of the IGH gene, as specified in the text. When we perform clonality analyses for diagnostic purpose we use to follow the Biomed2 guidelines (PMID: 22918122) which suggest to first look for a monoclonal IGH rearrangement and then, if polyclonal, to look for IGK or IGL gene rearrangements. The vertical barr of the graphs indicates the percentage of the total reads and for the IGK it corresponded to the 4.35% indicated in the table.

We agree with the Reviewer that without a formal proof, as a microdissection analysis, our observations would be only elusive, but unfortunately, the bioptic tissue has been extensively used for the immunohistochemical evaluation and molecular analyses. Therefore, in order to perform the microdissection analysis on H/RS-like cells we should have disrupted the sections on which the diagnosis has been made. We hope that our report would prompt other investigators to support our hypothesis.

POINT3: For case 3 there is unfortunately no graphical representation of the results, but a similar finding as for case 2, raising the same comment as for case 2.

RESPONSE 3: In figure 4 panels D-G refer to the IGK sequencing of case 3 as indicated in the text and figure legend.

POINT 4: In contrast to the histological/immunohistological description, the molecular findings were less precisely described and unclear to some extent. As already mentioned above, it is unclear what the “sequence”  in the tables for case 2 and 3 is representing. It is also not described which IgH framework region is amplified. A simple reference to the vendor is in this case not sufficient for a scientific publication.

RESPONSE 4: The sequences reported in the figure 4 are: for patient 2 the IGH framework region 1 (A-C) now indicated in the figure legend, and for patient 3 the Vk-Jk region and the Vk/Jk intron-Kde region already indicated in the figure legend.

POINT 5: Furthermore, the range of the amplicon sizes (IgH) appears to be very huge (203 to 306 bp).

RESPONSE 5: The amplicon size range 250-295 is that reported by the Invivoscribe gene scan fragment analysis for IGH FR2-JH primer sets (https://invivoscribe.com/uploads/products/instructionsForUse/280255.pdf) and by van Dongen JJ, et al. (Design and standardization of PCR primers and protocols for detection of clonal immunoglobulin and T-cell receptor gene recombinations in suspect lymphoproliferations: report of the BIOMED-2 Concerted Action BMH4-CT98-3936. Leukemia. 2003;17:2257-2317). The clonal peak found at 263bp in the two lesions of patient 1 falls within the reference range.

POINT 6: Why was IgL not performed for case 1; it should be able to support the IgH findings.

RESPONSE 6: The IGK rearrangement performed by gene scan analysis in case 1, now illustrated in supplementary figure 2 and added to the text, demonstrated the presence at diagnosis of a small clonal peak of the same size of the one detected in the relapse sample in both the master mixes used (i.e. 289bp and 284bp). This overlaps with what we obtained with the IGH rearrangement.

POINT 7: Finally, I assume that – based on length of the amplicons – somatic hypermutation (SHM) should be analyzable. For case 1, is SHM identical at both time points?  

RESPONSE 7: In case 1 the IGH FR2 sequence allowed for the analysis of the somatic hypermutation (SHM). Both samples were found to be highly hypermutated with 75% and 76% identity to the germline VH sequence. Moreover, they share the vast majority of the IGHV somatic mutations supporting a clonal relationship between the lesions. This has been now added to the text and showed in the supplementary figure 1.

Reviewer 3 Report

The authors report on 3 cases of primary cutaneous marginal zone B cell lymphoma containing CD30+ Hodgkin cells and focus on clonality assays through next generation sequencing methods.  The main finding of the paper is the finding of overall subclonality that appears consistent with the percentage of HRS cells in the lymphoid infiltrate in the biopsy.

I think the findings are interesting and potentially helpful to hematopathologists, although this is a small number of cases of a very rare entity.  NGS assay measurement of clonality in such cases is becoming more commonplace and this may be a useful reference for pathologists and guide clinicians to the diagnosis and monitoring of these entities.  The manuscript is overall well written.

My only suggestion prior to publishing would be pathology peer reviewer agreement that these cases do not seem to represent another entity such as classical Hodgkin lymphoma of the skin; EBV related CD30+ LPD of the skin does seem ruled out.  In the text I would also focus on CD30+ B cell cutaneous disorders as there is a good bit of background written about T cell lymphoma entities that is no consequential to this work.

Author Response

The authors report on 3 cases of primary cutaneous marginal zone B cell lymphoma containing CD30+ Hodgkin cells and focus on clonality assays through next generation sequencing methods.  The main finding of the paper is the finding of overall subclonality that appears consistent with the percentage of HRS cells in the lymphoid infiltrate in the biopsy.

I think the findings are interesting and potentially helpful to hematopathologists, although this is a small number of cases of a very rare entity.  NGS assay measurement of clonality in such cases is becoming more commonplace and this may be a useful reference for pathologists and guide clinicians to the diagnosis and monitoring of these entities.  The manuscript is overall well written.

POINT 1: My only suggestion prior to publishing would be pathology peer reviewer agreement that these cases do not seem to represent another entity such as classical Hodgkin lymphoma of the skin; EBV related CD30+ LPD of the skin does seem ruled out. 

RESPONSE 1: The cutaneous involvement by CHL has been mainly associated with nodal disease with a very few cases reported to be restricted to the skin. However, a diagnosis of primary cutaneous CHL was not favored in our cases because the background was not composed mainly by T cells, macrophages and eosinophils as in CHL but of B cells diffusely infiltrating the dermis or mainly organized in follicles with expanded marginal zones and monotypic plasmacells. In addition, the first case relapsed as a DLBCL, a transformation more commonly seen in nodular lymphocyte-predominant HL than in CHL. Similarly, a diagnosis of an EBV-related CD30+ lymphoproliferative disorder of the skin, such as a mucocutaneous ulcer, was ruled out by the negativity for EBER and the absence of iatrogenic immunosuppression. This comment has been now added to the discussion.

POINT 2: In the text I would also focus on CD30+ B cell cutaneous disorders as there is a good bit of background written about T cell lymphoma entities that is no consequential to this work.

RESPONSE 2: In the discussion we mentioned about the progression of LyP to ALCL because in the skin that was an example of a CD30+ disease, likely originated in the context of chronic inflammation, whose progression has been confirmed by clonality analysis. However, we agree with the Reviewer, and the sentence has now been re-formulated as follows: In the skin, similar molecular results have been reported in the possible progression of Lyp to cALCL [45-51] and of pseudo-B-cell-lymphomas to overt cutaneous B lymphomas [21-23]. In both situations, chronic inflammation has been suggested to play a major role by favoring the accumulation of somatic mutations in a clonal lymphoproliferative disorder.